# Exploring spatio-temporal impact of COVID-19 on citywide taxi demand: A case study of New York City

Yanan Zhang[1], Xueliang Sui[2], Shen Zhang[2]*

**1** School of Computer Science and Technology, Harbin University of Science and Technology, Harbin, Heilongjiang Province, China, **2** Department of Traffic Information and Control Engineering, Harbin Institute of Technology, Harbin, Heilongjiang Province, China

* shenzhang@hit.edu.cn

**Data Availability Statement:** Some data are available from the Figshare digital repository: https://figshare.com/s/de22839c51a1d81d4a12. Additional information is available at NYC OpenData (https://opendata.cityofnewyork.us).

## Abstract

Coronavirus disease 2019 (COVID-19) has brought dramatic changes in our daily life, especially in human mobility since 2020. As the major component of the integrated transport system in most cities, taxi trips represent a large portion of residents' urban mobility. Thus, quantifying the impacts of COVID-19 on city-wide taxi demand can help to better understand the reshaped travel patterns, optimize public-transport operational strategies, and gather emergency experience under the pressure of this pandemic. To achieve the objectives, the Geographically and Temporally Weighted Regression (GTWR) model is used to analyze the impact mechanism of COVID-19 on taxi demand in this study. City-wide taxi trip data from August 1st, 2020 to July 31st, 2021 in New York City was collected as model's dependent variables, and COVID-19 case rate, population density, road density, station density, points of interest (POI) were selected as the independent variables. By comparing GTWR model with traditional ordinary least square (OLS) model, temporally weighted regression model (TWR) and geographically weighted regression (GWR) model, a significantly better goodness of fit on spatial-temporal taxi data was observed for GTWR. Furthermore, temporal analysis, spatial analysis and the epidemic marginal effect were developed on the GTWR model results. The conclusions of this research are shown as follows: (1) The virus and health care become the major restraining and stimulative factors of taxi demand in post epidemic era. (2) The restraining level of COVID-19 on taxi demand is higher in cold weather. (3) The restraining level of COVID-19 on taxi demand is severely influenced by the curfew policy. (4) Although this virus decreases taxi demand in most of time and places, it can still increase taxi demand in some specific time and places. (5) Along with COVID-19, sports facilities and tourism become obstacles on increasing taxi demand in most of places and time in post epidemic era. The findings can provide useful insights for policymakers and stakeholders to improve the taxi operational efficiency during the remainder of the COVID-19 pandemic.

**Funding:** The author(s) received no specific funding for this work.

**Competing interests:** The authors have declared that no competing interests exist.

## 1. Introduction

Since 2020, the COVID-19 pandemic has spread globally with more than 279 million confirmed cases and 5.4 million deaths as of December 29, 2021. The majority of cases and deaths to date have been reported in the United States, India, Brazil and United Kingdom. The pandemic has undoubtedly brought enormous impacts and changes to economy, society, daily life and urban systems. As economic development progresses, urban transportation systems have been increasingly refined [1]. Concurrently, the outbreak and spread of the pandemic have precipitated unparalleled alterations in these transportation systems, a critical component of urban infrastructure, during the epidemic period. Human mobility, travel demand and trip pattern are being reshaped as well. Taking New York City (NYC) for instance, the decline rate of peak transit ridership reached to 92% in the week of April 6th, 2020 to April 12th, 2020, compared to the same period in 2019 (Wang et al. [2]). Apart from the decrease of vehicles on roads, this epidemic also led to fewer public transport demand (Paul et al. [3]), changes in trip modes and travel destination decisions (Simons et al. [4]). Therefore, it is necessary to consider COVID-19 as one of the influencing factors when exploring urban mobility and optimizing public transportation. Meanwhile, human mobility can be modelled as a mixture of transportation modes, while taxi is always one of the most important travel modes in most major cities. As a result, this paper attempts to investigate the impact of COVID-19 epidemic on the travel demand variation in post pandemic era, through a case study focusing on taxi services.

We focused on how COVID-19 influenced human mobility in New York City using a set of open data collected from Aug 1st, 2020, to July 31st, 2021. NYC was selected for the case study for several reasons, the most obvious of which is that NYC was typical, affected seriously by the epidemic and also one of the metropolises around the world. Another reason to choose NYC is the availability of data. NYC has been continuously tracking its entire taxi fleet since early 2009, while other kinds of related data is sufficient and easy to access (economy, population, GIS, epidemic, etc.).

Furthermore, Geographically and Temporally Weighted Regression (GTWR) approach is adopted to explore the spatio-temporal impact of COVID-19 on taxi demand. A total of 15 typical influencing factors were selected as the independent variables, including population density, COVID-19 case rate, etc. Using the principle of GTWR, a time-space taxi demand model was built based on 12 valid ones among these 15 independent variables. From time dimension and space dimension, these 12 types of explanatory variables' effects on taxi demand were analyzed. Concentrating on the effect of this outbreak, COVID-19's marginal effects on NYC's taxis were also analyzed spatially and temporally. At the end of our research, reasonable suggestions were given for this post-pandemic era.

There are two significances derived from this study, which are taxi demand analysis in post pandemic era and time-space effect of coronavirus on taxi demand respectively. By modeling the relationship between taxi demand and its influencing factors under the pressure of this epidemic, it become more reasonable to dispatch the whole city taxis, reducing the needless waiting time and consequent cost; and by studying this outbreak's spatiotemporal effect on taxi demand, combined with some rationalization proposals, some laws and experience about urban taxi operation on dealing with COVID-19 can be obtained, which will be useful when facing serious health emergencies next time in the future.

The rest of this paper contains aspects as follows: related representative studies about this pandemic's effect on transportation, and principle and application of GTWR model are shown in Section 2. Data source, model variables visualization and selection are demonstrated in Section 3. Section 4 introduced GTWR model and the generation of the taxi demand model. Temporal analysis, spatial analysis and marginal effect of COVID-19 on taxi demand are illustrated

in Section 5. According the model results and spatio-temporal analysis, some general laws and reasonable suggestions are obtained in Section 6 finally.

## 2. Literature review

Since the coronavirus outbreak, many researchers have correlated COVID-19 data with the traffic pattern variation and taken it as an influencing factor in urban transportation analyses. With the data from traffic counters, public transport ITS and traffic control cameras, Aloi et al. [5] revealed an urban mobility fall of 76%, and cities' public transport users dropped by up to 93%. Using official and secondary data in seven most populated cities in Colombia, Arellana et al. [6] found a distinct reduction of outdoor activities time and there was an about 30% increase in residential activities. In Chicago metropolitan area, Shamshiripour et al. [7] showed that 93% of respondents believed that transportation increased the risk of COVID-19 exposure.

Meanwhile, different models have been applied in analyzing the influence of COVID-19. Guo Y. et al. [8] proposed a Community Activity Score (CAS) and applied Negative Binomial models to model and forecast the spread of COVID-19 in Honolulu. Based on the Decision Tree approach, Pawar et al. [9] investigated the changes of traffic mode choices and found 5.3% of commuters shifted from public to private modes after the epidemic. With the Long Short-Term Memory Networks (LSTM), a type of recurrent neural networks (RNNs), Yao et al. [10] developed an accurate deep learning model and discovered that the daily confirmed COVID cases and daily fatality rate is highly related to the traffic volume in Detroit. Using a statistical method, i.e., bivariate analyses, Cusack [11] stated that nearly half of respondents changed their commute mode during the pandemic in Philadelphia. Chen et al. [12] quantified various factors (such as environmental, economic, social impacts, recycling modes) using an integrated framework supported by the analytical network process (ANP) and fuzzy comprehensive evaluation model. Zou et al. [13] introduced the CHMM method to analyze the interaction between driving behavior variables, which can better understand and explain driving behavior and original driving patterns, confirming the importance of considering the dependence between different variables. In order to overcome the uncertainty of the model, Wu et al. [14] applied the Bayesian model averaging (BMA) method to consider the advantages of different distributions in headway modeling. Zhang et al. [15] used the Hidden Markov Model with the Gaussian mixture model (GMM-HMM) approach to study the dynamic spatiotemporal characteristics and risk formation mechanism of vehicle lane changing.

Among various methods and diverse research themes in the analyses of COVID-19 influence on transportation, it is particularly meaningful to pay close attention to the COVID-19 data type, i.e., spatio-temporal data, because a confirmed COVID-19 case may influence transportation demand in its surrounding area and it could also affect the transportation pattern in several following days. Therefore, using temporal-spatial modeling methods is useful to analyze this effect. There are indeed some studies about the spatio-temporal analysis of COVID-19. Li S. et al. [16] analyzed spatiotemporal variation of air transportation influenced by the pandemic, and discovered the passenger throughput's changing rate is correlated to confirmed cases' growth. Saha et al. [17] explored the spatiotemporal variations in community mobility in India, and found the mobility towards residential area increased during the lockdown and decreased in unlocking period. Li A. et al. [18] focused on the virus' influence on micro-mobility, like bicycles, and found that activities towards home, park and grocery increased, while those towards leisure and shopping decreased, during lockdown period.

As for spatiotemporal taxi demand analysis, Liu Q. et al. [19] studied the relationship between urban environment and taxi demand, and pointed that taxi demand is high in densely

developed areas and more bus stops would reduce taxi demand. Tang et al. [20] used multi-community spatio-temporal graph convolutional network (MC_STGCN) to predict passenger demand in Shenzhen and New York City, and proved that MC_STGCN model had better performance than classical time-series model and deep learning model. Under the pressure of this pandemic, Yu et al. [21] employed multivariate linear mixed regression to disclose the relationship between taxi demand and COVID-19, and indicated that accumulated cured cases and blocking policy, have significantly influence on taxi demand in Ningbo, China. Zheng et al. [22] analyzed the variation of taxi market in 2019 lockdown and it was shown that even though taxi service reduced to some extent, it rebounded to exceed the pre-epidemic level after the lockdown. Considering that COVID-19 is kind of spatiotemporal data, using spatio temporal modeling methods may have better performance on illustrating the relationship between this pandemic and urban taxi demand and few studies used space-time modeling methods on this point. Therefore, a well-behaved temporal-spatial modeling method to fill in this gap is necessary, and we adopted GTWR model in this research.

## 3. Data and variables

### 3.1 Overview

From NYC Taxi and Limousine Commission (TLC), we got taxi trip data from Aug 1$^{st}$, 2020, to July 31$^{st}$, 2021. Combined with taxi zones data on NYC open data website, spatio-temporal taxi trip data was acquired. Also from NYC open data, we obtained NYC population distribution, zoning data and road network. Additionally, we collected COVID-19 case rate by MOD-ZCTA data on NYC health department, where MODZCTA is modified ZIP Code Tabulation Area geographies. Finally, bus stations, trains stations and Point of Interest (POI) data were achieved on Open Street Map.

### 3.2 Dependent variable

The number of pick-up taxies per taxi zone per hour was chosen as the dependent spatio-temporal variable to represent the taxi demand. In order to match weekly timescale COVID-19 data, we took the average of the hourly taxi demand over each week. There are 263 taxi zones in New York City and the sum of taxi demand per taxi zone in this one-year period is shown in Fig 1. It is clear that in central urban area, Manhattan, and in two airports, LGA & JFK, taxi demand is high. In Fig 2, we also plot the sum of taxi demand per hour in all taxi zones, and find that taxi demand is high in the afternoon but low at midnight.

### 3.3 Independent variable

Considering common influencing factors of taxi demand, population density, ratio of commercial area, road density, transportation stations, 11 types of POIs and COVID case rate were included as our independent variable.

**3.3.1 Population density.** Population density is a common variable when analyzing traffic demand. Since census tracts were different from taxi zones, we first divided the number of populations in each tract by its area and hence got corresponding population density. Next, we rasterized population density data in each census tract, and then each pixel got the value of corresponding population density. We then zonal summed up these pixels' values under the boundary of each taxi zone and got the overall value of each taxi zone. Finally, we divided each taxi zone's overall value by the number of pixels and got the population density in each taxi zone (see Fig 3).

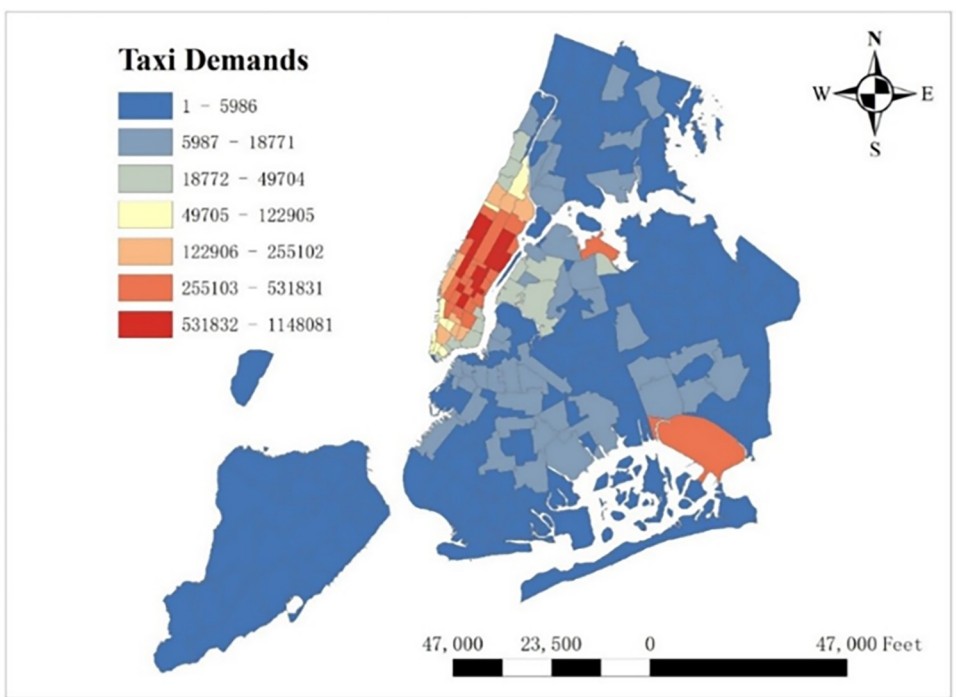

**Fig 1. Taxi demand per taxi zone in NYC.**

**3.3.2 Road density.** Road density can reflect the convenience of urban transportation to some extents and it also influences the demand of taxis. We calculate the road density per taxi zone as our explanatory variable whose unit is mile/mile$^2$ (see Fig 4).

**3.3.3 Transportation stations.** Considering there are kinds of public transportation, like buses, subways, railways influencing taxi demand, we first calculated the number of transportation stations per taxi zones. To be specific, this variable includes bus station, bus stop, railway

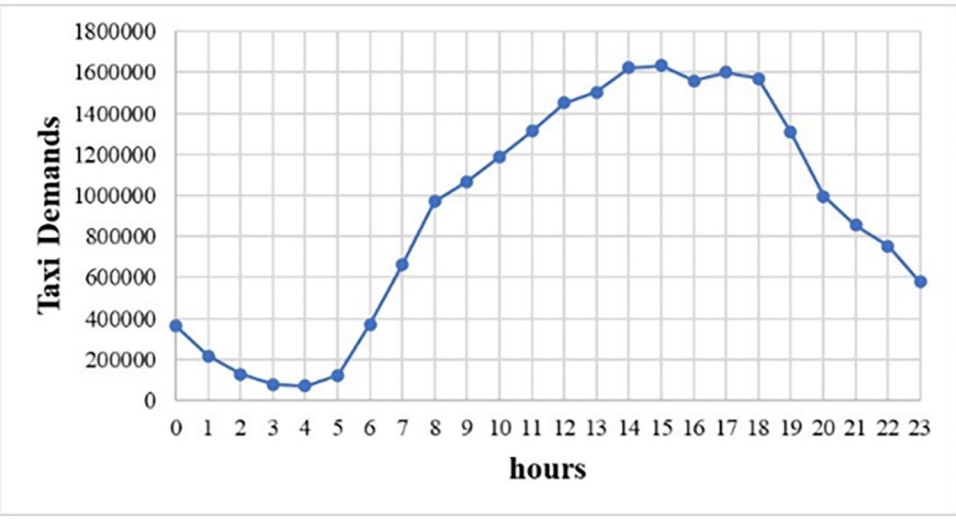

**Fig 2. Taxi demand per hour in NYC.**

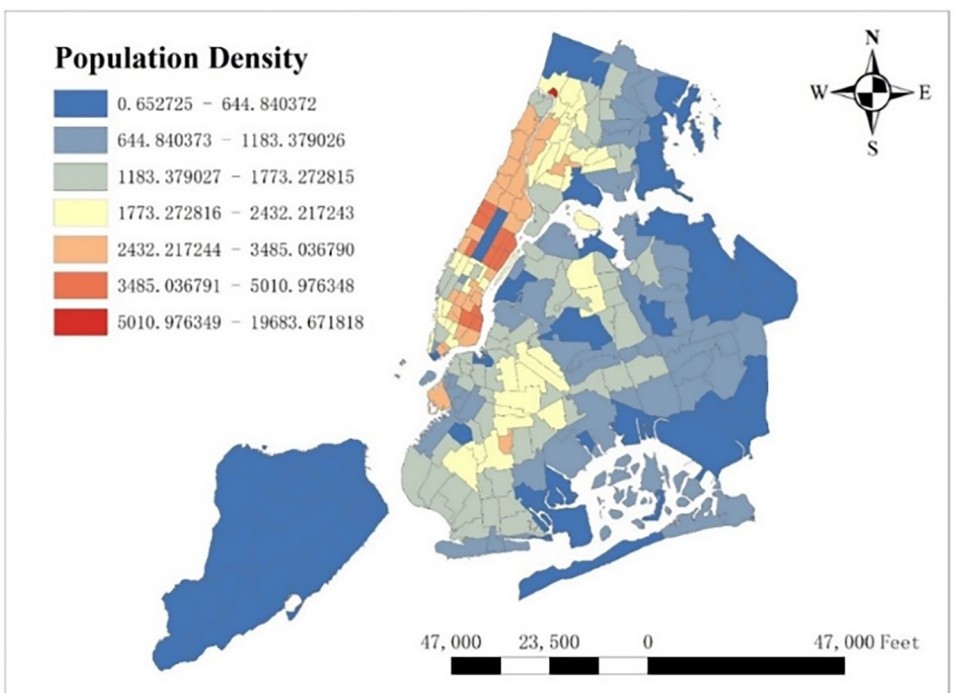

**Fig 3. Population density per taxi zone.**

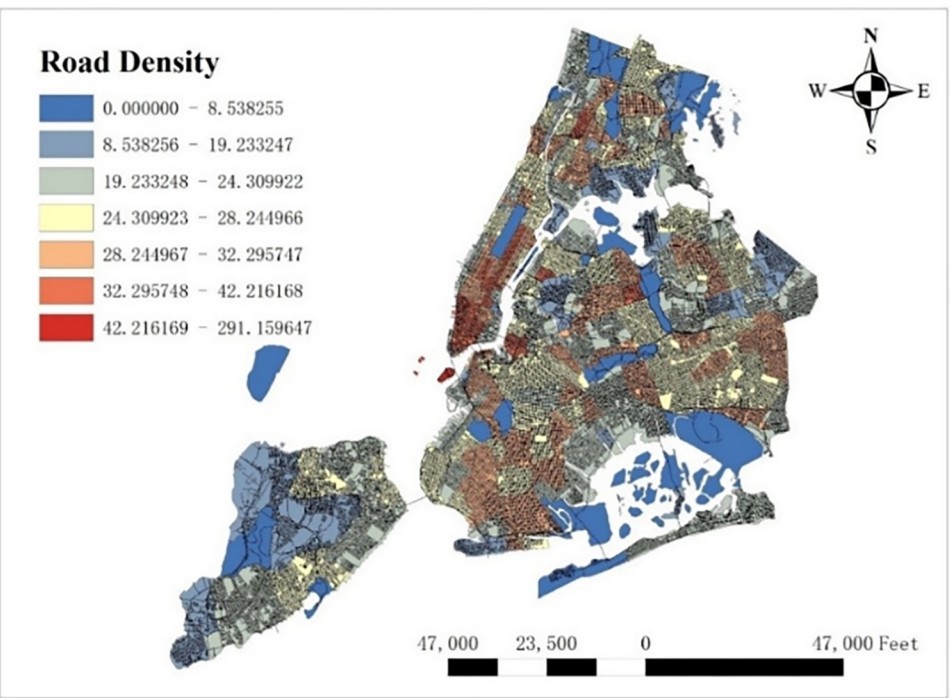

**Fig 4. Road density per taxi zone.**

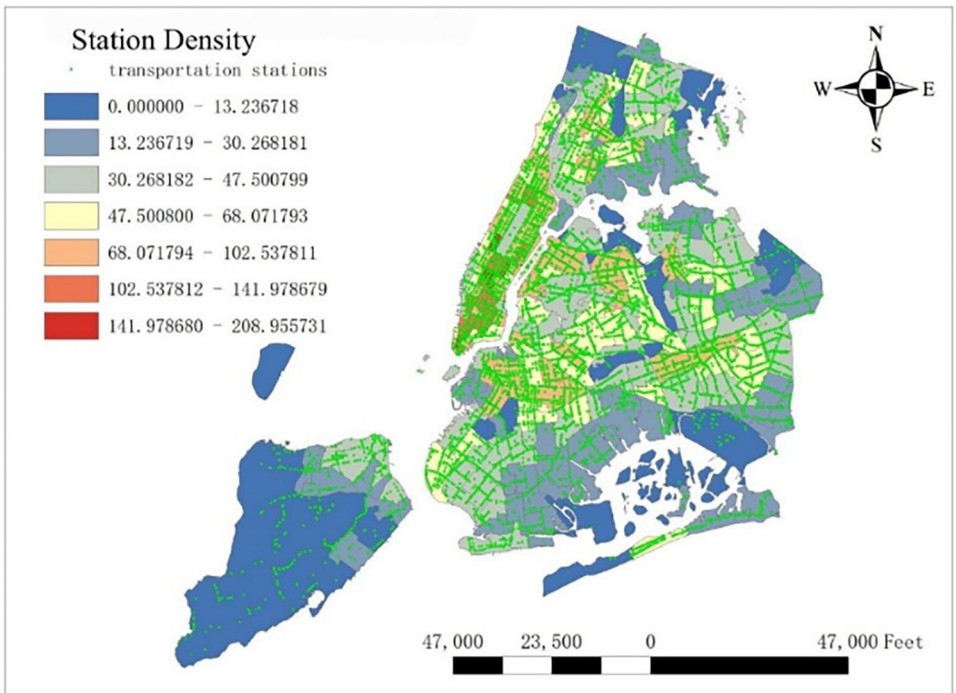

**Fig 5. Transportation station density in NYC.**

station and subway station. In order to eliminate the effect of area, we finally calculated the density of stations per taxi zone (see Fig 5), and the units are correspondingly number/mile$^2$.

**3.3.4 POIs.** Since point-of-interest (POI) data can be used to evaluate the land-use type, it is also used as an important factor in our analysis. We chose 11 types POIs from open street map, which is public, education, health, leisure, sport, catering, indoor accommodation, outdoor accommodation, shopping, money and tourism. The details are shown in Table 1. Like the station variable, we also calculated the density of POIs per taxi zone in order to eliminate the effect of area, and their units are number/mile$^2$.

**3.3.5 COVID-19 case rate.** COVID-19 case rate refers to the rate of cases per 100,000 people by week, which may influence the taxi demand in post-epidemic era. In Fig 6, case rate

**Table 1. Details of POIs.**

| Types | Details |
|---|---|
| Public | Providing public service, including police station, fire station, post office, etc. |
| Education | Universities, schools, kindergartens, etc. |
| Health | Hospitals, pharmacy, doctors, etc., which play important roles in post-epidemic era. |
| Leisure | Cinema, theatre, park, etc. |
| Sports | Swimming pool, tennis court, stadium, etc. |
| Catering | Restaurant, fast food, bar, café, etc. |
| Indoor accommodation | Hotel, motel, hostel, etc. |
| Outdoor accommodation | Shelter, campsite, etc. |
| Shopping | Supermarket, bakery, mall, etc. |
| Money | Bank, atm, etc. |
| Tourism | Museum, zoo, art, etc. |

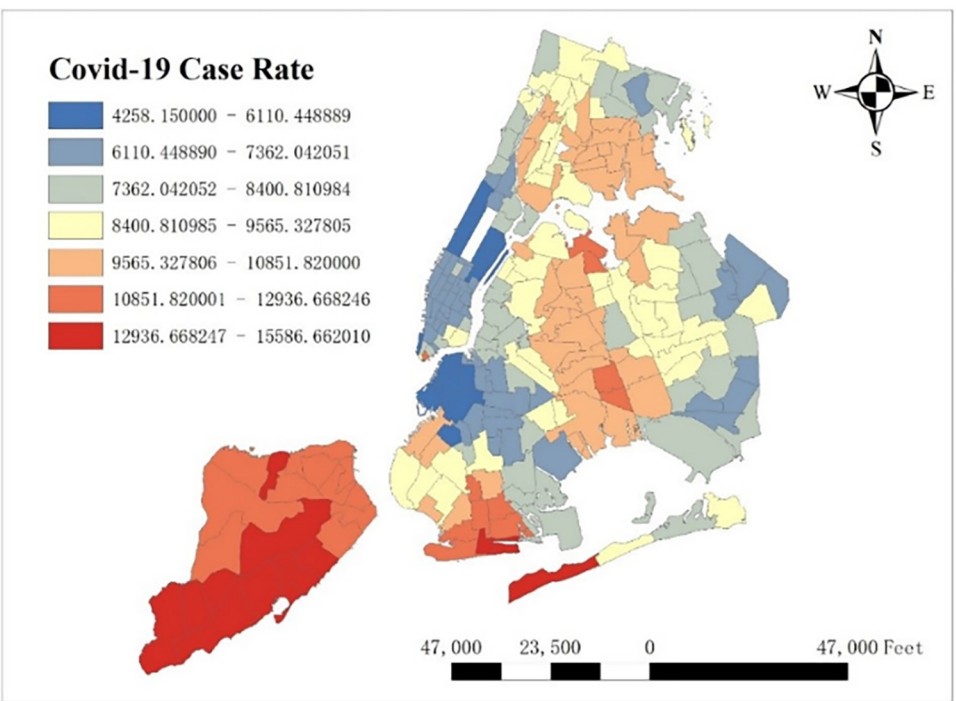

**Fig 6. Covid-19 case rate by taxi zones.**

distribution is shown and Staten Island has a high case rate compared with all NYC's boroughs. In Fig 7, we counted the case rate per week in all taxi zones and find case rate is high from week 18 (Nov. 29th, 2020 to Dec. 5th, 2020) to week 36 (Apr. 4th, 2021 to Apr. 10th, 2020), which means case rate is high in winter in New York City.

**3.4 Multicollinearity test.** Multicollinearity can influence the model accuracy. Here we calculated Pearson correlations among nearly all variables, except time-space COVID-19 variables. The result showed that catering was highly related to station, health, indoor accommodation, shop and finance; finance was highly related to indoor accommodation and shop; and shop was highly related to health. It was because the Pearson correlation coefficients between

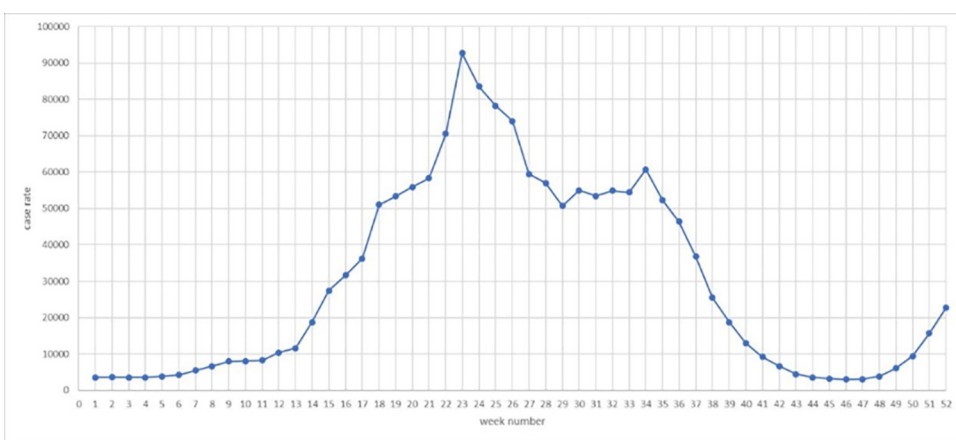

**Fig 7. Covid-19 case rate by taxi zones.**

**Table 2. Correlation coefficient matrix of remaining variables.**

|          | station  | public   | education | health   | leisure  | Sport    | indoor   | outdoor  | tour     | pop      | road     |
|----------|----------|----------|-----------|----------|----------|----------|----------|----------|----------|----------|----------|
| station  | 1        | 0.447652 | 0.496624  | 0.56988  | 0.456634 | 0.234101 | 0.580464 | 0.073449 | 0.424731 | 0.346554 | 0.37597  |
| public   | 0.447652 | 1        | 0.29287   | 0.46287  | 0.19958  | 0.153641 | 0.429238 | 0.108544 | 0.294709 | 0.210581 | 0.167501 |
| education| 0.496624 | 0.29287  | 1         | 0.574906 | 0.37469  | 0.236678 | 0.244779 | 0.154867 | 0.180813 | 0.337976 | 0.225188 |
| health   | 0.56988  | 0.46287  | 0.574906  | 1        | 0.332297 | 0.329808 | 0.477865 | 0.169332 | 0.284288 | 0.286522 | 0.250338 |
| leisure  | 0.456634 | 0.19958  | 0.37469   | 0.332297 | 1        | 0.234172 | 0.40329  | 0.144457 | 0.38881  | 0.244426 | 0.122112 |
| Sport    | 0.234101 | 0.153641 | 0.236678  | 0.329808 | 0.234172 | 1        | 0.384113 | 0.150668 | 0.258586 | 0.098993 | 0.018718 |
| indoor   | 0.580464 | 0.429238 | 0.244779  | 0.477865 | 0.40329  | 0.384113 | 1        | 0.016229 | 0.390702 | 0.177038 | 0.193957 |
| outdoor  | 0.073449 | 0.108544 | 0.154867  | 0.169332 | 0.144457 | 0.150668 | 0.016229 | 1        | 0.063994 | 0.078776 | 0.010787 |
| tour     | 0.424731 | 0.294709 | 0.180813  | 0.284288 | 0.38881  | 0.258586 | 0.390702 | 0.063994 | 1        | 0.078027 | 0.069748 |
| pop      | 0.346554 | 0.210581 | 0.337976  | 0.286522 | 0.244426 | 0.098993 | 0.177038 | 0.078776 | 0.078027 | 1        | 0.12362  |
| road     | 0.37597  | 0.167501 | 0.225188  | 0.250338 | 0.122112 | 0.018718 | 0.193957 | 0.010787 | 0.069748 | 0.12362  | 1        |

these variables were larger than 0.7. Finally, catering, finance, shop variables were deleted. Table 2 is the correlation coefficient matrix of remaining variables.

## 4. Methodology

### 4.1 GTWR model

By extending Geographically Weighted Regression (GWR) model with temporality, Huang et al. [23] developed GTWR model and applied it on real spatio-temporal data. Dong et al. [24] built the GTWR model to analyze determinants of haze pollution in China, and found that economic development and industry upgrading are main solutions to reduce haze pollution, while transportation industry and construction industry are main sources of haze pollution. Based on GTWR model, Liu J. et al. [25] found carbon emission intensity in China were influenced by urbanization, population, etc., and energy intensity had positive impact on carbon emission intensity. With the help of GTWR's principle, Guo B. et al. [26] explored the effects of socioeconomic and environment on chronic obstructive pulmonary disease mortality, and found the influence degree of anthropic factors are higher than natural factors.

With usually high goodness of fit and clear explanation of temporal-spatial characteristics, GTWR was also widely used in transportation analysis. Shen et al. [27] adopted GTWR modeling principle to analyze how land use and household properties influenced automobile travel demand, and their high accuracy study showed the influence of above two kinds of factors on travel demand varies temporally and spatially. In order to explore pedestrian injury severity geographically and temporally in Hong Kong, Xu et al. [28] used GTWR and found that it was significantly influenced by vehicles number, speed limit, injury location, etc. Ma et al. [29] utilized GTWR to model the relationship between bike-sharing usage and its determinants, and found that the elderly proportion and entertainment density have different correlation between dock less bike-sharing and docked bike-sharing. Based on GTWR, Ma et al. [30] revealed the relationship between built environment and transit ridership, and stated that temporal heterogeneity of coefficients was the key determinant of transit ridership per TAZ.

With the help of GTWR model, we built a spatio-temporal regression model and spatial-temporal distribution of coefficients generated by the model was used to analyze how different factors influence taxi demand in both time dimension and space dimension.

Generally, GTWR model can be formulated as follows:

$$y_i = \beta_0(u_i, v_i, t_i) + \sum_{k=1}^{d} \beta_k(u_i, v_i, t_i)x_{ik} + \varepsilon_i, \; i = 1, 2, \ldots, n \tag{1}$$

where $y_i$; $x_{i1}, x_{i2}, \ldots, x_{im}$ are dependent variable $y$ and independent variable $x_{i1}, x_{i2}, \ldots, x_{im}$ at time $t_i$ at point $(u_i, v_i)$. $\beta_k(u_i, v_i, t_i)$ is the regression coefficient at time $t_i$ at point $(u_i, v_i)$, $\beta_0(u_i, v_i, t_i)$ is the intercept at time $t_i$ at point or grid $(u_i, v_i)$ and $\varepsilon_i$ is the error term at $i$ whose mean is supposed to 0 and variance is supposed to $\sigma^2$.

The estimation of GTWR model is shown as follows:

$$\hat{\beta}(u_i, v_i, t_i) = [X'W(u_i, v_i, t_i)X]^{-1} X'W(u_i, v_i, t_i)Y \qquad (2)$$

where $W(u_i, v_i, t_i)$ is space-time weighted matrix like:

$$W(u_i, v_i, t_i) = \begin{bmatrix} w_1(u_i, v_i, t_i) & 0 & \cdots & 0 \\ 0 & w_2(u_i, v_i, t_i) & \cdots & 0 \\ \vdots & \vdots & \ddots & \vdots \\ 0 & 0 & \cdots & w_n(u_i, v_i, t_i) \end{bmatrix} \qquad (3)$$

and $w_j(u_i, v_i, t_i) = w_{ij}$ can be calculate by Gaussian, Bi-square and Exponential kernel functions, which respectively are,

$$w_{ij}^{Gaussian} = exp\left[-\frac{(d_{ij}^{ST})^2}{h_{ST}^{Gaussian}2}\right] \qquad (4)$$

$$w_{ij}^{Bi-square} = \begin{cases} \left[1 - \frac{(d_{ij}^{ST})^2}{h_{ST}^{Bi-square}2}\right]^2, & d_{ij} \leq h_{ST}^{Bi-square} \\ 0, & d_{ij} > h_{ST}^{Bi-square} \end{cases} \qquad (5)$$

$$w_{ij}^{Exponential} = exp\left[-\frac{d_{ij}^{ST}}{h_{ST}^{Exponential}}\right] \qquad (6)$$

where $d_{ij}^{ST}$ is spatial-temporal distance between position $i$ and $j$ and $h_{ST}$ is Space-time bandwidth parameter. We can calculate $d_{ij}^{ST}$ by Eq (7) as below.

$$d_{ij}^{ST} = \sqrt{\lambda[(u_i - u_j)^2 + (v_i - v_j)^2] + \mu(t_i - t_j)^2} \qquad (7)$$

where $\lambda$ and $\mu$ are scale factors used to balance the effects of temporal and spatial distances. When $\lambda = 0$, GTWR model becomes temporal weighted regression (TWR) model, which only considers the time effect; And when $\mu = 0$, GTWR model becomes geographically weighted regression (GWR) model, which only considers the spatio effect.

## 4.2 Data modeling

Based on the principle of GTWR model, our taxi demand analysis model was formulated as Eq (8).

$$\begin{aligned} Freq_i = & \beta_0(u_i, v_i, t_i) + \beta_1(u_i, v_i, t_i)STA_i + \beta_2(u_i, v_i, t_i)PUB_i + \beta_3(u_i, v_i, t_i)EDU_i \\ & + \beta_4(u_i, v_i, t_i)HEA_i + \beta_5(u_i, v_i, t_i)LEI_i + \beta_6(u_i, v_i, t_i)SPO_i + \beta_7(u_i, v_i, t_i)IND_i \\ & + \beta_8(u_i, v_i, t_i)OUT_i + \beta_9(u_i, v_i, t_i)TOU_i + \beta_{10}(u_i, v_i, t_i)POP_i + \beta_{11}(u_i, v_i, t_i)ROA_i \\ & + \beta_{12}(u_i, v_i, t_i)COV_i + \varepsilon_i, \ i \\ & = 1, 2, \ldots, n \end{aligned} \qquad (8)$$

**Table 3. Components of GTWR model.**

|  | Label | Units | Details | Mean | Std. | Median |
|---|---|---|---|---|---|---|
| **freq** | **Freq** | **counts** | **Dependent variable** | **12.91** | **32.42** | **1.50** |
| station | STA | counts / mile$^2$ | Independent variable | 61.72 | 36.17 | 58.00 |
| public | PUB | counts / mile$^2$ | Independent variable | 23.68 | 43.03 | 9.23 |
| education | EDU | counts / mile$^2$ | Independent variable | 7.65 | 9.06 | 4.31 |
| health | HEA | counts / mile$^2$ | Independent variable | 10.41 | 13.41 | 6.11 |
| leisure | LEI | counts / mile$^2$ | Independent variable | 6.08 | 9.12 | 3.14 |
| Sport | SPO | counts / mile$^2$ | Independent variable | 1.50 | 3.72 | 0.00 |
| indoor | IND | counts / mile$^2$ | Independent variable | 4.10 | 10.18 | 0.00 |
| outdoor | OUT | counts / mile$^2$ | Independent variable | 0.52 | 3.38 | 0.00 |
| tour | TOU | counts / mile$^2$ | Independent variable | 12.12 | 27.95 | 1.71 |
| pop | POP | mile / mile$^2$ | Independent variable | 1806.00 | 1366.52 | 1528.04 |
| road | ROA | foot / mile$^2$ | Independent variable | 37.26 | 56.11 | 28.00 |
| covid | COV | case / 100000 | Independent variable | 156.43 | 152.05 | 94.24 |
| X | X | miles | X coordinate | 190.64 | 3.93 | 189.99 |
| Y | Y | miles | Y coordinate | 39.45 | 4.98 | 39.66 |
| time | t | - | timestamp | - | - | - |

where $\beta_0 (u_i, v_i, t_i)$ is the intercept at time $i$ at taxi zone $i$, $\beta_i (u_i, v_i, t_i)$ is the regression coefficient at time $i$ at taxi zone $i$, and $\varepsilon_i$ is the error term.

Table 3 describes one dependent variable (see *freq*), 12 independent variables, X, Y coordinates, time stamp, and their details. Besides, we calculated the X, Y coordinates of each taxi zone's centroid and recorded timestamps.

From the data description in Table 3, we found that there are too many "0" in data which may influence our model accuracy. In order to eliminate the effect of "0", we normalized our dependent variables using Eq (9).

$$x' = \frac{x - \mu}{\sigma} \tag{9}$$

where $x'$ is new data, $x$ is original data, $\mu$ is the mean of the sample data set, and $\sigma$ is the std. of the sample data set.

## 5. Results

### 5.1 Model estimation result

As shown in Table 4, we compared the model performance based on OLS, GWR, TWR and GTWR. Considering time effect and space effect, GWR, TWR and GTWR have better model performance than traditional OLS. In addition, while GWR only concentrates on space effect and TWR only concentrates on time effect, GTWR model combines the time-space effect as a whole, which performs better on spatio-temporal data theoretically than the other two models.

**Table 4. Model comparison.**

| Model | R$^2$ | AIC |
|---|---|---|
| OLS | 0.264 | 2341551 |
| TWR | 0.303 | 2328311 |
| GWR | 0.515 | 2239429 |
| GTWR | 0.814 | 2035823 |

**Table 5. Estimation results of GTWR model.**

| Explanatory variable | Average Coefficient | Std. of Coefficient | Min Coefficient | Median Coefficient | Max Coefficient |
|---|---|---|---|---|---|
| Intercept | 13.327 | 35.728 | -198.302 | 10.118 | 561.480 |
| station | 1.412 | 5.043 | -31.620 | 0.802 | 24.953 |
| public | 4.051 | 7.904 | -85.980 | 4.997 | 19.155 |
| education | 3.282 | 4.037 | -19.940 | 2.144 | 74.218 |
| health | 4.888 | 6.744 | -8.625 | 2.251 | 46.452 |
| leisure | 0.837 | 3.883 | -34.118 | 0.503 | 53.152 |
| sport | -1.232 | 2.469 | -35.441 | -0.874 | 16.107 |
| indoor | 4.550 | 4.501 | -19.418 | 3.729 | 42.488 |
| outdoor | 2.373 | 7.562 | -6.545 | 0.056 | 80.365 |
| tour | -1.643 | 6.153 | -68.268 | -1.085 | 48.913 |
| pop | 3.981 | 6.556 | -48.103 | 3.315 | 52.188 |
| road | -2.440 | 8.723 | -92.853 | -0.515 | 54.646 |
| covid | -5.742 | 39.495 | -279.518 | -3.604 | 566.615 |

The model estimation result proves this conjecture. Since GTWR's $R^2$ is the closest to 1 and its AIC is minimum, it can be known that GTWR model has the best goodness-of-fit among these four models.

The result of GTWR model is shown in Table 5. From the sign of average coefficients and median coefficients of 12 types of dependent variables, station, public, education, health, leisure, indoor accommodation, outdoor accommodation and population are positive, while sport, tourism, road density and COVID-19 case rate is negative, indicating that in post-epidemic era, stations, public facilities, education, health facilities, leisure facilities and accommodation can increase taxi demand, and other factors may reduce the demand. Among these influencing factors' average coefficients, the sign of COVID-19 cases rate's mean coefficient is negative, and the absolute value is the largest, which shows that COVID-19 has been a major factor in reducing the taxi demand in post pandemic era. Corresponding to COVID-19, health facilities, like hospital and clinic, shows a positive sign of average coefficient and its absolute value is relatively large, illustrating health facilities has become one of the major factors increasing taxi demand. The possible reason is that under the pandemic, people try not to go to places with high COVID cases, and their awareness of medical treatment increases.

## 5.2 Spatial feature of coefficients

With GTWR model and geographic information system (GIS), spatial distribution of some important coefficients can be visualized. By calculating the average coefficients of different dependent variables in this one-year time period, which are shown from Figs 8–15, it could be found that the promotion and inhibition degree of each explanatory variable varies from zone to zone.

Station density increases taxi demand in Lower Manhattan and northern Brooklyn, but decreases it near JFK Airport. Public facilities density increases taxi demand in most of New York City, especially in Bronx, but decreases it near the Southwest of Central Park. Education facilities density attracts taxis in most of Manhattan and Queens, but has a negative effect on taxi demand in most of Brooklyn and Bronx. Health facilities density promotes taxi demand in most of NYC, especially in Upper Manhattan and southern Bronx. Leisure facilities reduce taxi demand around Central Park and JFK Airport, but increase it in the Staten Island and southwest of Brooklyn. Sport facilities decrease taxi demand in most of NYC, except for the

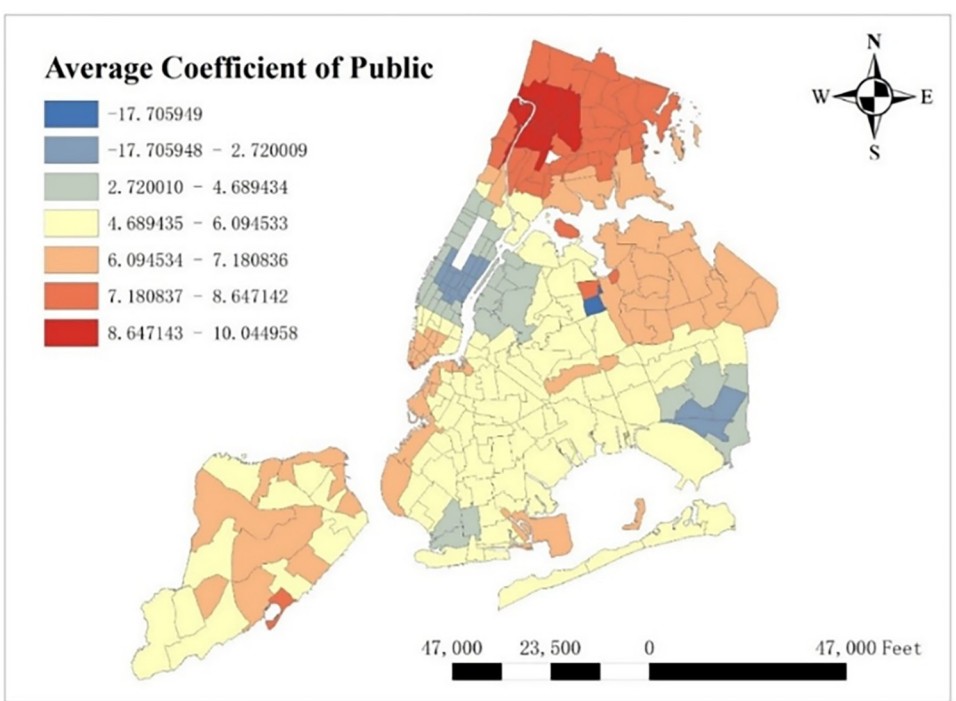

**Fig 8. Spatial distribution of public's average coefficient.**

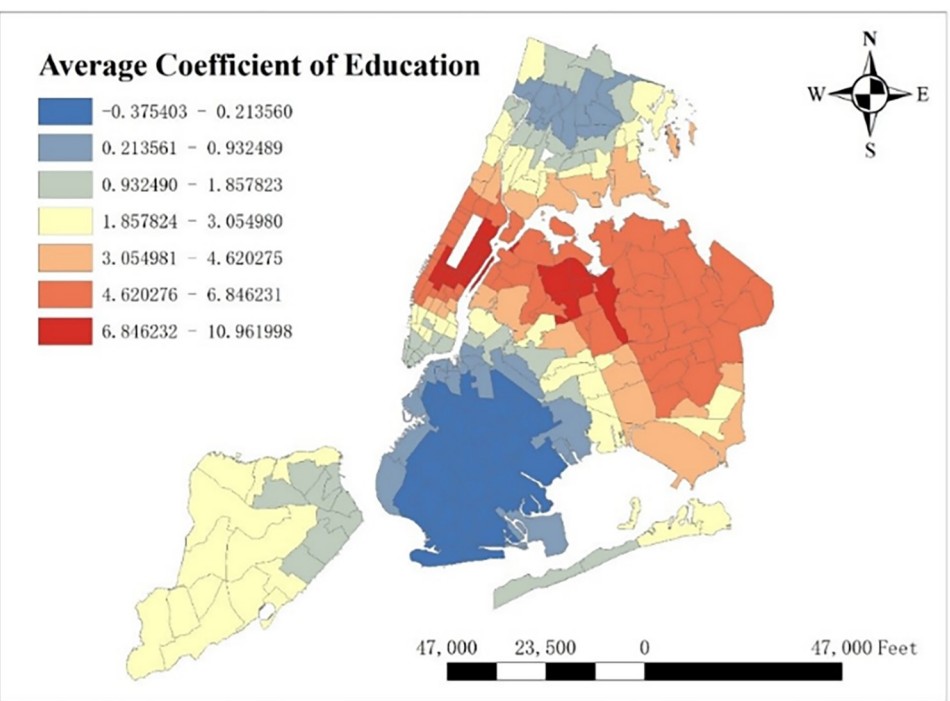

**Fig 9. Spatial distribution of education's average coefficient.**

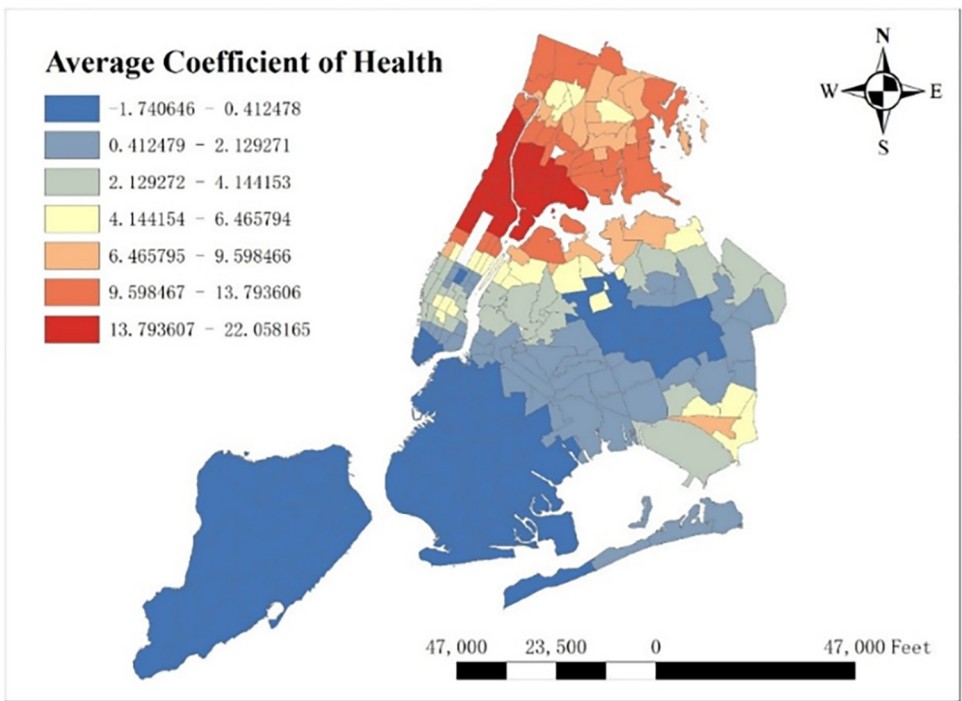

**Fig 10. Spatial distribution of health's average coefficient.**

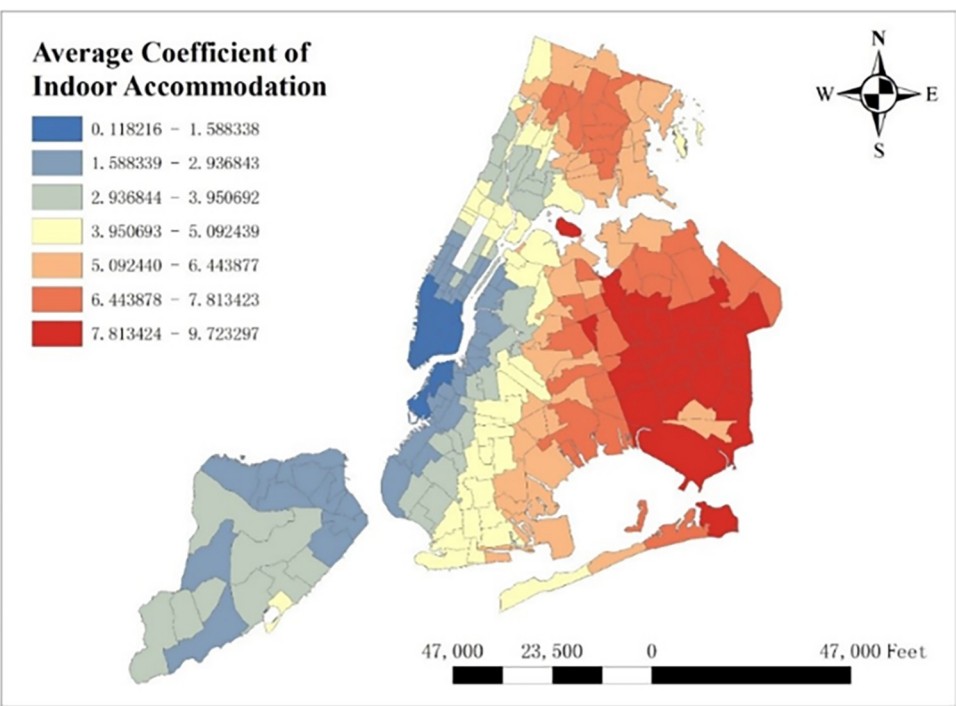

**Fig 11. Spatial distribution of indoor accommodation's average coefficient.**

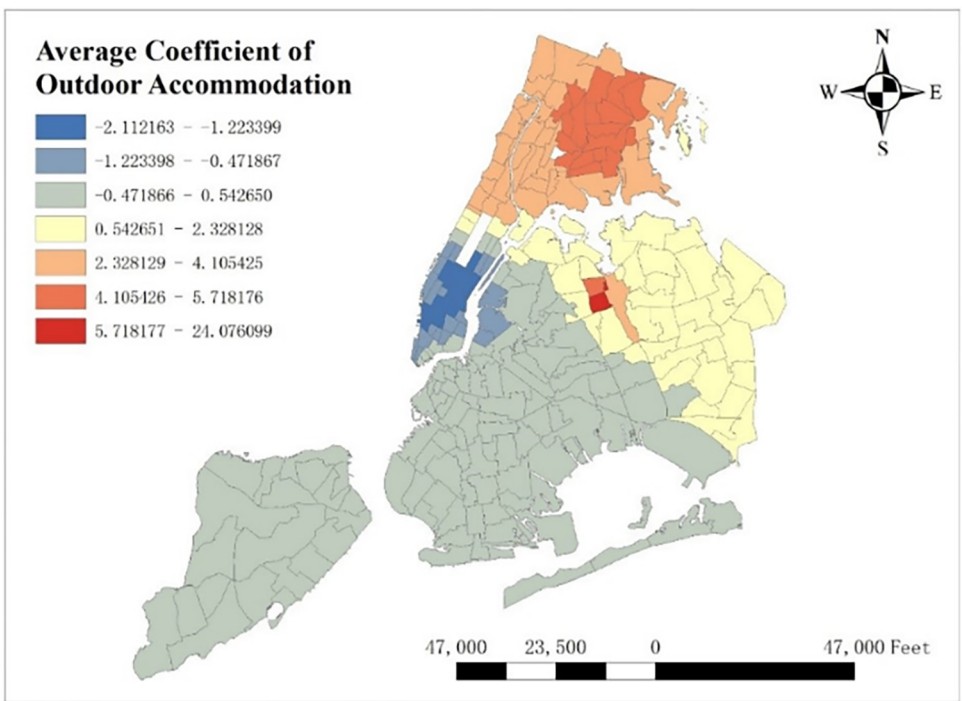

**Fig 12. Spatial distribution of outdoor accommodation's average coefficient.**

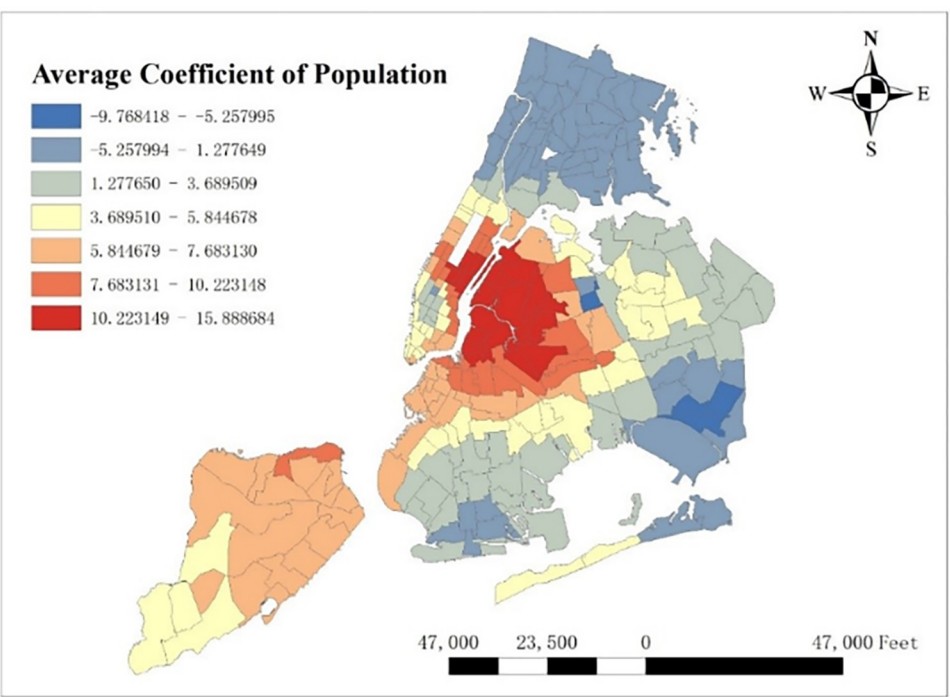

**Fig 13. Spatial distribution of population density's average coefficient.**

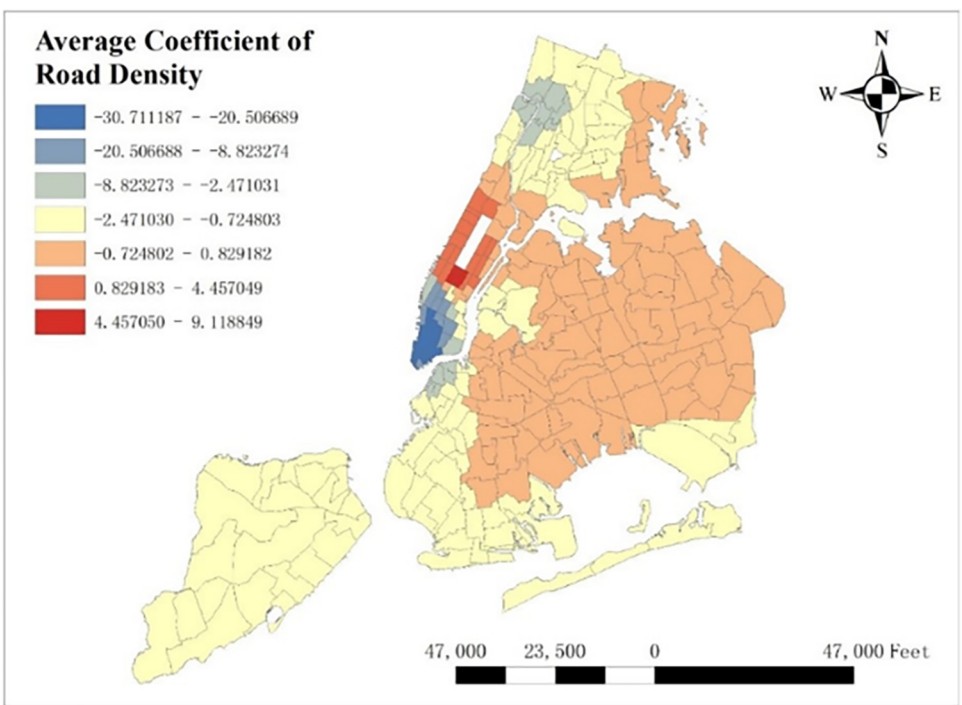

**Fig 14. Spatial distribution of road density's average coefficient.**

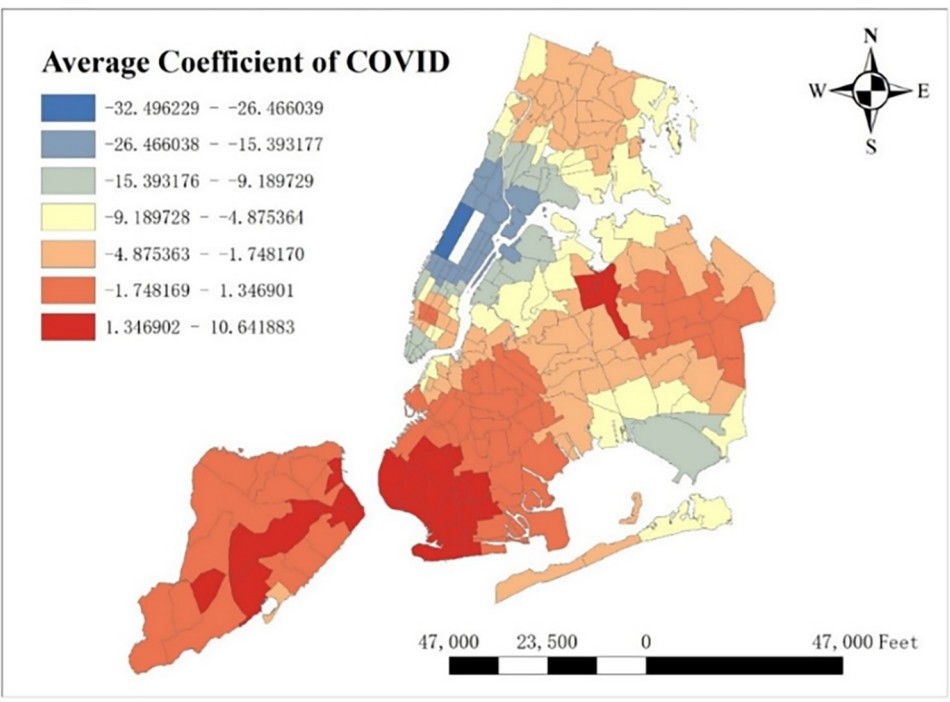

**Fig 15. Spatial distribution of covid's average coefficient.**

places around JFK Airport. Tourism boosts taxi demand around Central Park and JFK Airport but decreases it in Lower Manhattan. Population density attracts taxi demand in northern Brooklyn, western Queens and around Central Park, but decreases it in most of Bronx and JFK Airport. Road density factor boosts taxi demand near Central Park especially around Times Square, but decreases it in Lower Manhattan.

As for COVID-19, it reduces taxi demand in most of NYC, except in southwest Brooklyn and some parts of Staten Island. In addition, the reducing degree in area around the Central Park is larger than other areas, and its possible reason is that in central NYC, the epidemic prevention policy is stricter than other area, leading to high level of inhibiting effect on taxi demand.

## 5.3 Temporal feature of coefficients

The results of GTWR can also provide us the temporal changes of the average coefficients. As shown in Fig 16, in terms of daily average regression coefficients, with the largest negative average coefficients, COVID-19 has become the major inhibiting factor of taxi demand in post era; in contrast to this, because of the largest positive coefficients, health has become the major promoting factor of taxi demand. Therefore, under the pandemic pressure, health care and epidemic have become the major influencing factor.

From 0:00 to 24:00, It is shown that COVID-19's coefficients climb up to the maximum from 0:00 to 6:00, then decrease to the minimum from 6:00 to 20:00, and increases from 20:00 to 24:00. Especially when it is 8pm, the average coefficient reaches the minimum, indicating that COVID-19 has the largest reducing degree to taxi demand; also, COVID coefficients at night is much lower than it in daytime, illustrating that the reducing degree at night is much larger than it in daytime. The possible reason of this phenomenon is Curfew policy beginning at 20:00. On the contrary, the coefficients value of health is always the largest, and its changing scope is fewer that COVID-19's. Especially in the afternoon, it reaches the maximum, causing the largest attractive level of taxi demand.

Except for COVID-19, tourism and sport always share negative coefficients, leading to the decrease in taxi demand. It seems to be inconsistent with common sense that some people may tend to take taxis after tiring tour or exercises. Under the pressure of this pandemic, we

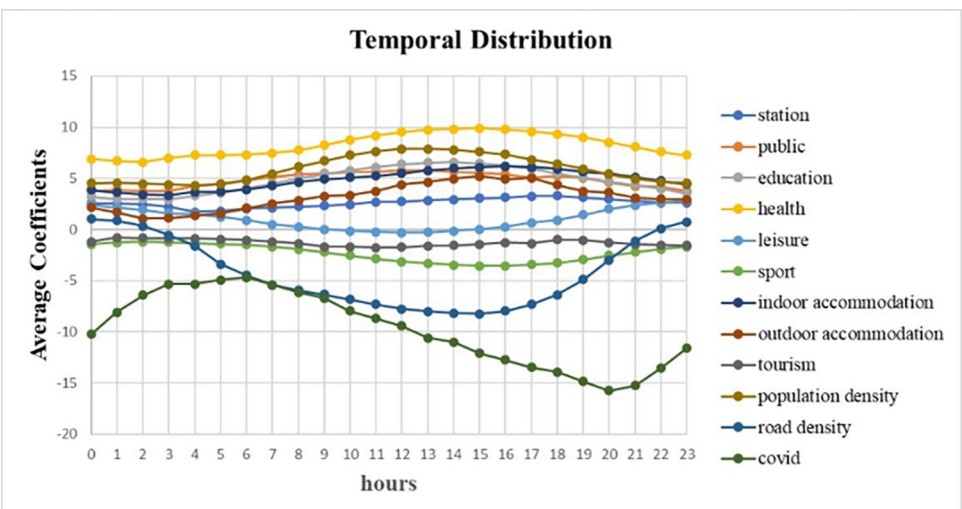

**Fig 16. Temporal distribution of average coefficients.**

suppose the potential reason is that the virus makes the tourist area or sports center partially closed, and then these areas no longer attract taxis. Another negative factor is road density and it indeed decrease the taxi demand in most of daytime. Road density can reflect the convenience of local transportation and there will be more kinds and quantity of vehicles, like more buses, more subways and more private cars, in high road density area. These vehicles act as the competitors of taxis and hence road density become one of the inhibiting factors.

Among the promoting factors, most of their coefficients increase at 8:00 and decrease at night, showing a scenario that people begin their daily life at 8am and go home for rest at night. However, leisure factor is on the contrary of this trend: it has less attraction to taxi demand in daytime, but begin to increase its attracting level from 18:00. It can be used to explain this phenomenon that people need to work in daytime from 8:00 to 16:00, when leisure's coefficient is closer to 0, and they enjoy their night life after that. The further inference is that people will still go to leisure area, even though there are Curfew policy and covid virus.

## 5.4 Marginal spatial-temporal effect of COVID-19 on taxi demand

**5.4.1 Seasonal effect.** From Fig 7, we know that COVID-19 case rate is high in winter. In order to explore the seasonal effect of COVID-19 on taxi demand, we scale down average taxi demand, average covid case and average covid coefficients per week, and plot them in Fig 17. From week 0 to week 11, week 13 to week 41, week 45 and week 48, when it is Aug 1st to Oct 24th, Nov 11th to May 22nd, Jun 13th to Jun 19th and Jul 4th to Jul 10th, the sign of coefficients is negative, especially in week 4 and 10. But in week 12, week 42 to 44, week 46 to 47 and week 49 to 51, when it is Oct 25th to Oct 31st, May 23rd to Jun 12th, Jun 20th to Jul 3rd, Jul 11th to Jul 31st, the sign of coefficients is positive, indicating that covid-19 actually increase the taxi demand. Therefore, it is clear that in most time of autumn, winter, spring, COVID-19 depresses the taxi demand, but in some time of summer COVID-19 actually boosts taxi demand. The potential reason can be attributable to the characteristics of COVID-19 virus. Since people are more susceptible to infection in winter, they share higher awareness to the virus and avoid going to high case rate places in cold days. Moreover, they may also prefer other travel modes to public transportation in winter. Furthermore, some time-variant restraining order may also reduce the taxi demand in those high case rate places.

However, it is irreconcilable with our common sense, which is widely believed that COVID-19 should depress city-wide taxi demand in both summer and winter. In order to reveal the deep reason of this phenomenon, we extract the spatial distribution of average COVID-19's coefficients in week 47 (From Jun 20th, 2021 to Jun 26th, 2021) per taxi zones,

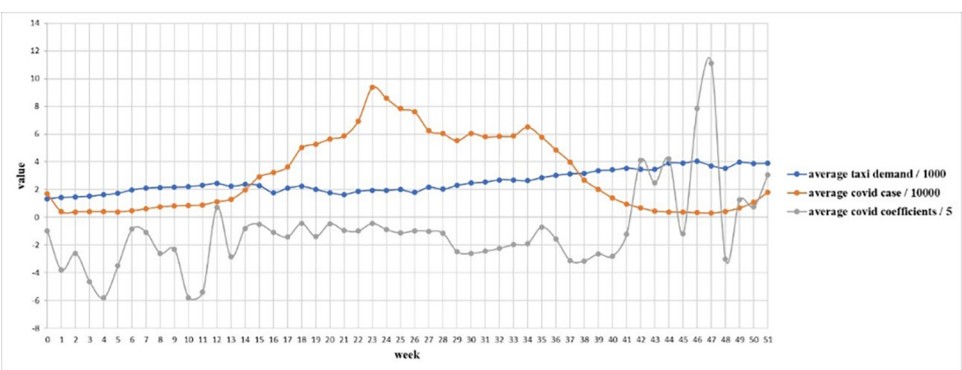

**Fig 17. Changes of average taxi demand, case rate and coefficients by week.**

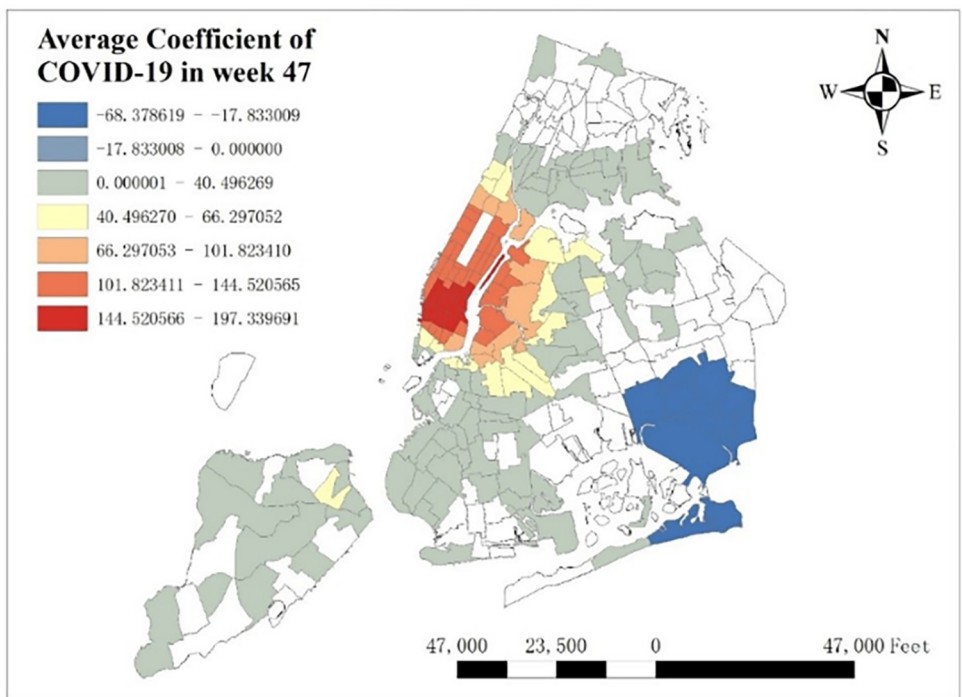

**Fig 18. Average COVID-19 coefficient in week47.**

which has the highest average positive coefficient temporally shown in Fig 18. One thing to note is that we removed the coefficients with low significance, whose absolute t-value is less than 2. To make clear comparison, average taxi demand per zone in week 47 is shown in Fig 19. It is shown that the highest COVID-19 coefficients focus around in Lower Manhattan and the lowest coefficients concentrate around JFK Airport, while taxi demand in these two regions is relatively high. The potential reason of this difference is that as an international airport, JFK Airport holds strictly epidemic prevention and control policy to prevent the spread of this pandemic, so COVID-19 has much negative influence on taxi demand. However, in Lower Manhattan and as shown in Fig 6, lower case rate doesn't reduce taxi demand but makes contribution to it.

**5.4.2 Rush hour effect.** From Fig 16, we know daily trend of COVID-19's coefficients and find that its average coefficients decrease from 6:00 to 20:00 and increase from 20:00 to 5:00. Besides, it should be more meaningful to explore how COVID-19 influences taxi demand in rush hours. Here we select 6:00 to 10:00 as morning rush hours and 15:00 to 19:00 as evening rush hours. The spatial distribution of average COVID-19 coefficients in both morning rush hours and evening rush hours are shown in Figs 20 and 21.

From Figs 20 and 21, global spatial distribution of average COVID-19 coefficients shows that COVID-19 depresses taxi demand in most of NYC, especially in areas around the Central Park and around JFK and LGA Airport, while it induces taxi demand in most of Staten Island, southwest Brooklyn and central Queens. Along with general spatial distribution, it will also be meaningful to find some differences between morning and evening rush hours, and there indeed is an obvious change in lower Manhattan. In morning rush hours, lower Manhattan is the only place in Manhattan that attracts taxi demand, while it decreases taxi demand in evening rush hour.

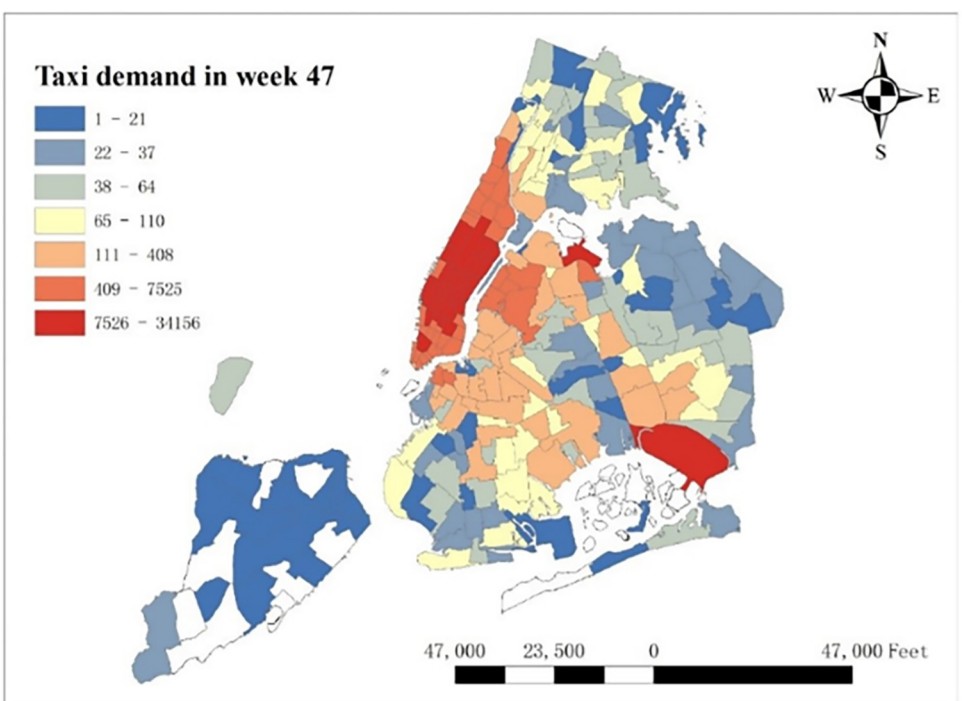

**Fig 19. Average taxi demand in week47.**

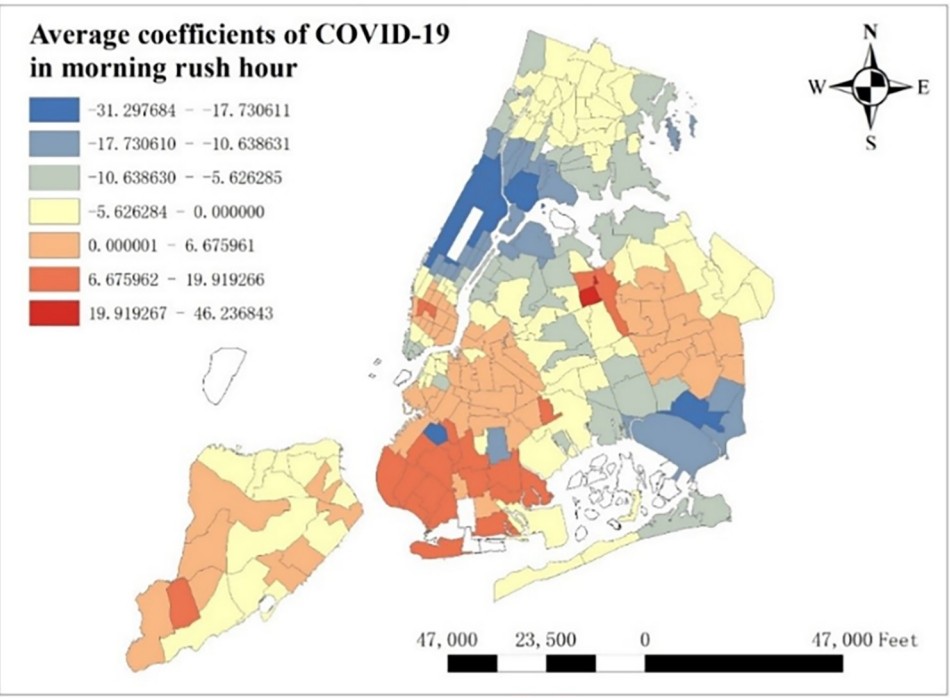

**Fig 20. Average coefficients of COVID-19 in morning rush hour.**

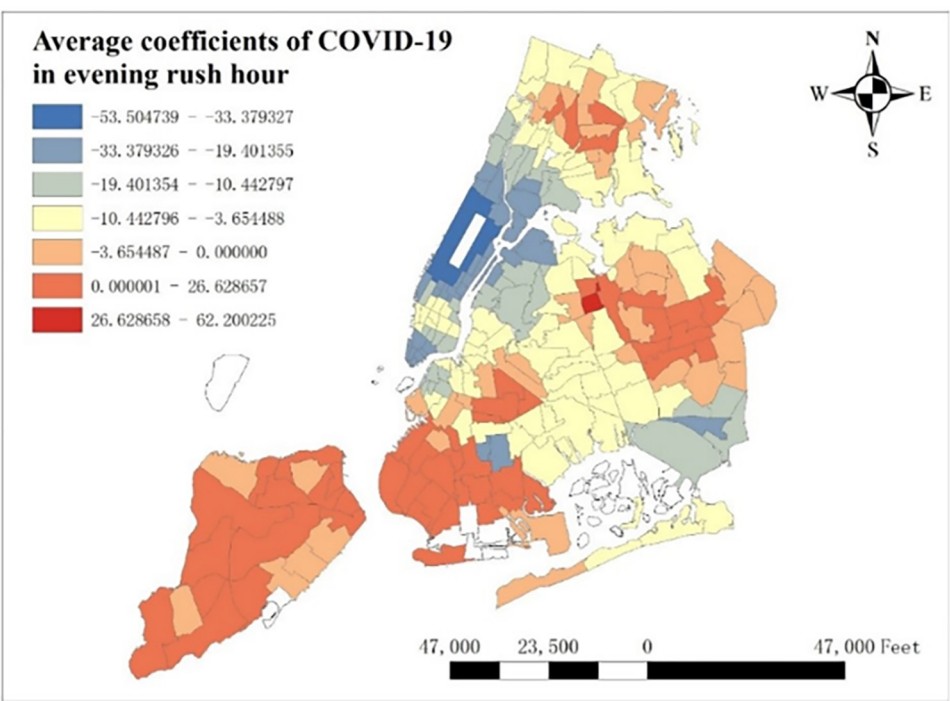

**Fig 21. Average coefficients of COVID-19 in evening rush hour.**

## 6. Conclusion

This research quantitatively analyzes the impacts of the pandemic COVID-19 on taxi demand temporally and spatially in New York City. After multicollinearity test, we selected 12 types of explanatory variables, including COVID-19 case rate data. Using GTWR modeling principle, a taxi demand analysis model with $R^2$ of 0.814 and AIC of 2035823 was built, which perform better than traditional OLS, TWR and GWR models. Then, we analyzed this pandemic's spatial-temporal effect on taxi demand from Aug 1st, 2020 to Jul 31st, 2021 in all NYC's taxi zones.

Major findings of this research can be concluded as: (1) GTWR has greater fitting performance on spatial-temporal data than traditional OLS, TWR and GWR models (2) COVID-19 and health care become the leading inhibiting and promoting factors of taxi demand in post epidemic era. (3) The inhibitory degree of this pandemic on taxi demand is larger in cold weather than it is in hot weather. (4) The inhibitory degree of this pandemic on taxi demand is severely influenced by the curfew policy and it reaches the maximum at the beginning of curfew at 20:00. (5) Although this virus dampens taxi demand in most of time and places, it still promotes taxi demand in some specific time and places. (6) Sports and tourism also become obstructive factors on the increase of taxi demand in most of places and time in post epidemic era. With these conclusions, GTWR model is verified to be effective and workable on space-time impact analysis of the epidemic; and it is suggested that taxis drivers avoid going places with high virus case rate in winter; and more taxis should be dispatched around health center, while fewer taxis should be around tourism and sports area in this post epidemic era. Additionally, the reinforcement of public health infrastructure is crucial, including enhancing the emergency response capabilities of medical systems and improving disease monitoring and prevention. Embracing remote working and digital transformation is essential for adapting to the post-epidemic work and lifestyle changes. This transformation encompasses fostering the technical capabilities of both public and private sectors. Furthermore, the promotion of digital

payment methods to minimize physical contact is recommended, alongside advancing the digitization of taxi dispatch and tracking systems to elevate service efficiency and enhance passenger experience. Finally, collaboration with technology providers, government agencies, and other transportation services is essential to explore innovative service models and business strategies, addressing the evolving market demands of the post-epidemic era. Moreover, under the pressure of this pandemic, it can be optimistic that COVID-19 doesn't reduce all taxi demand, but actually boosts it in some specific area and time.

Further research can focus on following fields: (1) Comparing GTWR model with other spatio-temporal models, like Bayesian hierarchy model and Recurrent Neural Network (RNN), and analyzing the advantages of GTWR model. (2) Using GTWR model to analyze this pandemic's effect on taxi demand in other cities and making contrastive analysis between this effect in NYC and other cities. (3) more transportation entities, like buses and subways, can be included to extend this research's application scenario. (4) The article uses New York City as an example to conduct GTWR analysis. The applicability to other cities needs to be analyzed. The universality of the method can be proved through comparisons in various cities.

## Author Contributions

**Conceptualization:** Yanan Zhang, Shen Zhang.

**Data curation:** Yanan Zhang, Shen Zhang.

**Formal analysis:** Shen Zhang.

**Investigation:** Shen Zhang.

**Methodology:** Yanan Zhang, Xueliang Sui, Shen Zhang.

**Validation:** Xueliang Sui.

**Writing – review & editing:** Xueliang Sui.

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
