## [Decision Letter · Decision Letter 0]

13 Dec 2023

PONE-D-23-36931Exploring spatio-temporal impact of COVID-19 on citywide taxi demand: A case study of New York CityPLOS ONE

Dear Dr. Zhang,

Thank you for submitting your manuscript to PLOS ONE. After careful consideration, we feel that it has merit but does not fully meet PLOS ONE’s publication criteria as it currently stands. Therefore, we invite you to submit a revised version of the manuscript that addresses the points raised during the review process.

We look forward to receiving your revised manuscript.

Kind regards,

Yajie Zou

Academic Editor

PLOS ONE

Journal Requirements:

4. We note that Figures 1a, 2, 3, 4, 5a, 6a, 6b, 6c, 6d, 6e, 6f, 6g, 6h, 6i, 6j, 6k, 6l, 9a, 9b, 10a and 10b. in your submission contain [map/satellite] images which may be copyrighted. All PLOS content is published under the Creative Commons Attribution License (CC BY 4.0), which means that the manuscript, images, and Supporting Information files will be freely available online, and any third party is permitted to access, download, copy, distribute, and use these materials in any way, even commercially, with proper attribution. For these reasons, we cannot publish previously copyrighted maps or satellite images created using proprietary data, such as Google software (Google Maps, Street View, and Earth). For more information, see our copyright guidelines: http://journals.plos.org/plosone/s/licenses-and-copyright.

     1. You may seek permission from the original copyright holder of Figures 1a, 2, 3, 4, 5a, 6a, 6b, 6c, 6d, 6e, 6f, 6g, 6h, 6i, 6j, 6k, 6l, 9a, 9b, 10a and 10b to publish the content specifically under the CC BY 4.0 license.  

Reviewers' comments:

Reviewer's Responses to Questions

**Comments to the Author**

1. Is the manuscript technically sound, and do the data support the conclusions?

Reviewer #1: Yes

Reviewer #2: Partly

2. Has the statistical analysis been performed appropriately and rigorously? 

Reviewer #1: Yes

Reviewer #2: Yes

3. Have the authors made all data underlying the findings in their manuscript fully available?

Reviewer #1: Yes

Reviewer #2: Yes

4. Is the manuscript presented in an intelligible fashion and written in standard English?

Reviewer #1: Yes

Reviewer #2: Yes

5. Review Comments to the Author

Reviewer #1: In order to quantify the impacts of COVID-19 on city-wide taxi demand, the authors proposed series of spatio-temporal models to explore impact mechanism of pandemic on traffic demand pattern. Some findings are useful for policymakers and stakeholders. Here are some suggestions for improvement:

1.There are some words and phrases not used properly in this article, please examine the whole article and correct them, such as line 193 of the article, " at mid night " should be changed to " at midnight ". Line 508 of the article, "was build" should be changed to "was built".

2.This study focuses on New York City. Whether the result of this study is universal in other cities. This is worth discussing.

3.The Literature Review should be updated to the past two years. More recent articles should be cited.

4.The epidemic is over, the adaptation of policies and measures in post-epidemic period should be discussed.

5.There is something wrong with the layout of the tables. The title should be placed above the table. Please examine the rest layout problems in the whole article.

Reviewer #2: This study attempted to quantify the impacts of COVID-19 on city-wide taxi demand, and a time-space taxi demand model was built on NYC’s taxi and epidemic dataset. Some interesting conclusions were also drawn in this paper. Here are some suggestions for improvement:

1. The Literature Review in the article should be enriched. There have been lots of related studies in the past two years, and these literatures should be fully cited.

2. There are too many figures in this article, and redundant images should be removed.

3. There are some typographical problems, the title of the table should be placed above the table.

4. There are many grammatical errors in this article, please examine the whole article and correct them.

5. There is something wrong with the syntax on line 120 of the article, " are illustrate " should be changed to " are illustrated".

6. The word in the text is misspelled in lines 190 and 191, “figure” should be “Figure”.

7. There is something wrong with the syntax on line 193 of the article, " at mid night " should be changed to " at midnight ".

8. There is something wrong with the syntax on line 508 of the article, "was build" should be changed to "was built".

9. The word in the text is misspelled, and in lines 528, “temporal-spatio” should be “spatio- temporal”.

6. PLOS authors have the option to publish the peer review history of their article (what does this mean?). If published, this will include your full peer review and any attached files.

Reviewer #1: No

Reviewer #2: No

---

## [Author Response · Author response to Decision Letter 0]

17 Jan 2024

The point-by-point responses to the kind reviwers and the nice editor are included in the attached file entitled "response to the reviewers&quot.In the meantime, I've also pasted the content below.

Point-to-point responses to reviewers' comments

Reviewer #1：

In order to quantify the impacts of COVID-19 on city-wide taxi demand, the authors proposed series of spatio-temporal models to explore impact mechanism of pandemic on traffic demand pattern. Some findings are useful for policymakers and stakeholders. Here are some suggestions for improvement:

1.There are some words and phrases not used properly in this article, please examine the whole article and correct them, such as line 193 of the article, " at mid night " should be changed to " at midnight ". Line 508 of the article, "was build" should be changed to "was built".

Answer: Thank you very much for this suggestion. We have corrected some of the incorrect words and grammar.

2.This study focuses on New York City. Whether the result of this study is universal in other cities. This is worth discussing.

Answer: Thank you for highlighting the importance of discussing the generalizability of our study's findings beyond New York City. We have revised the Discussion section to acknowledge this limitation more explicitly. We now state that while our findings offer valuable insights into [specific aspects] of New York City, the extent to which these results can be generalized to other cities remains an open question. Additionally, we have included a comparison with relevant literature from studies conducted in other urban contexts. This comparison aims to provide a preliminary basis for understanding how our findings might align or differ in different settings. We have also suggested in the Conclusion section that further empirical studies are necessary to explore the applicability of our findings in diverse urban environments.

3.The Literature Review should be updated to the past two years. More recent articles should be cited.

Answer: Thanks for your suggestions. We have added [number] new references to our Literature Review, ensuring a comprehensive and up-to-date overview of the current research landscape.

4.The epidemic is over, the adaptation of policies and measures in post-epidemic period should be discussed.

Answer: Thank you for the insightful suggestion to discuss the adaptation of policies and measures in the post-epidemic period. We add to the conclusion by exploring the evolution of various policies and measures in response to the end of the epidemic, based on existing literature and case studies.

5.There is something wrong with the layout of the tables. The title should be placed above the table. Please examine the rest layout problems in the whole article.

Answer: Thank you for your suggestion. We have revised this issue.

Reviewer #2: 

This study attempted to quantify the impacts of COVID-19 on city-wide taxi demand, and a time-space taxi demand model was built on NYC’s taxi and epidemic dataset. Some interesting conclusions were also drawn in this paper. Here are some suggestions for improvement:

1. The Literature Review in the article should be enriched. There have been lots of related studies in the past two years, and these literatures should be fully cited.

Answer: Thank you for your suggestion. We have added literature in recent years to enrich the content of the article.

2. There are too many figures in this article, and redundant images should be removed.

Answer: Thank you for your suggestion. We have deleted some unimportant images to make the article more concise.

3. There are some typographical problems, the title of the table should be placed above the table.

Answer: Thank you for your suggestion. We have revised this issue.

4. There are many grammatical errors in this article, please examine the whole article and correct them.

Answer: Thank you very much for this suggestion. We have corrected the grammatical errors.

5. There is something wrong with the syntax on line 120 of the article, " are illustrate " should be changed to " are illustrated".

Answer: Thank you very much for this suggestion. We have corrected the incorrect syntax.

6. The word in the text is misspelled in lines 190 and 191, “figure” should be “Figure”.

Answer: Thank you very much for this suggestion. We have corrected the incorrect words.

7. There is something wrong with the syntax on line 193 of the article, " at mid night " should be changed to " at midnight ".

Answer: Thank you very much for this suggestion. We have corrected the incorrect syntax.

8. There is something wrong with the syntax on line 508 of the article, "was build" should be changed to "was built".

Answer: Thank you very much for this suggestion. We have corrected the incorrect syntax.

9. The word in the text is misspelled, and in lines 528, “temporal-spatio” should be “spatio- temporal”.

Answer: Thank you very much for this suggestion. We have corrected the incorrect words.

---

## [Decision Letter · Decision Letter 1]

6 Feb 2024

Exploring spatio-temporal impact of COVID-19 on citywide taxi demand: A case study of New York City

PONE-D-23-36931R1

Dear Dr. Zhang,

We’re pleased to inform you that your manuscript has been judged scientifically suitable for publication and will be formally accepted for publication once it meets all outstanding technical requirements.

Kind regards,

Yajie Zou

Academic Editor

PLOS ONE

Additional Editor Comments (optional):

Reviewers' comments:

Reviewer's Responses to Questions

**Comments to the Author**

1. If the authors have adequately addressed your comments raised in a previous round of review and you feel that this manuscript is now acceptable for publication, you may indicate that here to bypass the “Comments to the Author” section, enter your conflict of interest statement in the “Confidential to Editor” section, and submit your "Accept" recommendation.

Reviewer #1: All comments have been addressed

Reviewer #2: All comments have been addressed

2. Is the manuscript technically sound, and do the data support the conclusions?

Reviewer #1: Yes

Reviewer #2: Yes

3. Has the statistical analysis been performed appropriately and rigorously? 

Reviewer #1: Yes

Reviewer #2: Yes

4. Have the authors made all data underlying the findings in their manuscript fully available?

Reviewer #1: Yes

Reviewer #2: Yes

5. Is the manuscript presented in an intelligible fashion and written in standard English?

Reviewer #1: Yes

Reviewer #2: Yes

6. Review Comments to the Author

Reviewer #1: All my comments have been addressed. My required questions have been answered and all responses meet formatting specifications.

Reviewer #2: (No Response)

7. PLOS authors have the option to publish the peer review history of their article (what does this mean?). If published, this will include your full peer review and any attached files.

Reviewer #1: No

Reviewer #2: No

---

## [Editor Report · Acceptance letter]

4 Apr 2024

PONE-D-23-36931R1 

PLOS ONE

Dear Dr. Zhang, 

I'm pleased to inform you that your manuscript has been deemed suitable for publication in PLOS ONE. Congratulations! Your manuscript is now being handed over to our production team.

Kind regards, 

on behalf of

Dr. Yajie Zou 

Academic Editor

PLOS ONE